# Construction of Recombinant *Magnetospirillum* Strains for Nitrate Removal from Wastewater Based on Magnetic Adsorption

Haolan Zheng, Bo Pang, Shuli Li, Shijiao Ma, Junjie Xu, Ying Wen and Jiesheng Tian *

State Key Laboratory of Agrobiotechnology, College of Biological Sciences, China Agricultural University, Beijing 100193, China; haolanzheng@cau.edu.cn (H.Z.); caupb@cau.com (B.P.); lishuler@163.com (S.L.); msj_91@cau.edu.cn (S.M.); junjiexue89@163.com (J.X.); dwyt2756@cau.edu.cn (Y.W.)
* Correspondence: tianhome@cau.edu.cn

**Abstract:** Nitrate ion ($NO_3^-$) in wastewater is a major cause of pollution in aquatic environments worldwide. *Magnetospirillum gryphiswaldense* (MSR-1) has a complete dissimilatory denitrification pathway, converts $NO_3^-$ in water into nitrogen ($N_2$) and simultaneously removes ammonium ions ($NH_4^+$). We investigated and confirmed direct effects of regulatory protein factors Mg2046 and MgFnr on MSR-1 denitrification pathway by EMSAs and ChIP-qPCR assays. Corresponding mutant strains were constructed. Denitrification efficiency in synthetic wastewater medium during a 12-h cell growth period was significantly higher for mutant strain $\Delta mgfnr$ (0.456 mmol·$L^{-1}$·$h^{-1}$) than for wild-type (0.362 mmol·$L^{-1}$·$h^{-1}$). Presence of magnetic particles (magnetosomes) in MSR-1 greatly facilitates collection and isolation of bacterial cells (and activated sludge) by addition of a magnetic field. The easy separation of magnetotactic bacteria, such as MSR-1 and $\Delta mgfnr$, from wastewater using magnetic fields is a unique feature that makes them promising candidates for practical application in wastewater treatment and sludge pretreatment.

**Keywords:** *Magnetospirillum gryphiswaldense*; magnetic adsorption; dissimilatory denitrification pathway; nitrogen removal; wastewater treatment



## 1. Introduction

Nitrate ($NO_3^-$) pollution in water and wastewater is a major environmental problem worldwide. The primary sources of $NO_3^-$ contamination are anthropogenic activities such as excessive fertilization in agriculture [1]. In numerous lakes and rivers in Europe and the U.S., $NO_3^-$ concentration has doubled in less than a decade [2]. In certain coastal zones (e.g., Gulf of Mexico, Yellow Sea, Baltic Sea, Chesapeake Bay) increasing $NO_3^-$ levels cause severe algal blooms that lead to "dead zones". $NO_3^-$ can infiltrate drinking water sources, posing a threat to human health. A 2020 report from the Ireland Environmental Protection Agency [3] concludes that "urgent" action is needed to reduce water $NO_3^-$ levels.

Among various approaches for $NO_3^-$ removal, biological denitrification is more cost-effective than physicochemical methods, such as adsorption, ion exchange, and reverse osmosis [4,5]. Biological denitrification has been applied widely for treatment of municipal, industrial, and agriculture wastewater and $NO_3^-$-contaminated groundwater. During this process, $NO_3^-$ functions as an electron acceptor and is converted to harmless nitrogen gas by denitrifying microorganisms [6–10]. An electron donor is necessary as a source of electrons and energy during denitrification processes [11].

Denitrification is classified as heterotrophic or autotrophic, depending on the electron donor. In heterotrophic denitrification, the electron donor is a low-molecular-weight (e.g., acetate, methanol, glucose, benzene, methane) or high-molecular-weight (e.g., cellulose, polylactic acid, polycaprolactone) organic compound [11]. In autotrophic denitrification, the electron donor is an inorganic compound such as hydrogen gas ($H_2$), reduced sulfur

compound (e.g., sulfide, elemental sulfur, thiosulfate), ferrous iron ($Fe^{2+}$), iron sulfide (e.g., FeS, $Fe_{1-x}S$, $FeS_2$), arsenite ($As^{III}$), or manganese ($Mn^{II}$) [12]. The nitrate removal rate is generally higher for heterotrophic denitrification than for autotrophic denitrification because heterotrophic bacteria have a faster growth rate [4].

Many studies during the past decade have focused on denitrification processes involving $Fe^{2+}$ as an electron donor. Most $Fe^{II}$-oxidizing denitrifiers isolated to date are heterotrophic and can grow only in the presence of organic carbon [13,14]. Effective follow-up treatments for excess sludge from heterotrophic denitrification are needed [15]. Following long-term operations, accumulated $Fe^{2+}$ may form an iron crust that inhibits subsequent activity of denitrifiers.

Magnetotactic bacteria (MTB) produce intracellular magnetic nanoparticles composed of magnetite ($Fe_3O_4$) or greigite ($Fe_3S_4$) and enclosed by a lipid bilayer, termed "magnetosomes" [16]. These nanosized organelles enable MTB to orient and migrate along magnetic lines of force. MTB have been applied in cancer therapy and in heavy metal biosorption from wastewater or polluted water [17,18]. They are useful for nitrate removal from wastewater because MTB biomass is easily removed from sewage treatment systems by magnetic separation. Li et al. [19] described a complete denitrification pathway, including functional genes for reduction of nitrate (*nap*), nitrite (*nir*), nitric oxide (*nor*), and nitrous oxide (*nos*), in the extensively studied MTB strain *Magnetospirillum gryphiswaldense* MSR-1. The present study is focused on regulation of the denitrification pathway in MSR-1, and construction of a recombinant MTB strain with enhanced nitrate removal capability.

## 2. Materials and Methods

### 2.1. Bacterial Strains and Culture Conditions

Bacterial strains and plasmids used in this study are listed in Table 1. MSR-1 was autoclaved and cultured at 30 °C in sodium lactate medium (SLM), which contained (per L) 2.25 g of sodium lactate solution (55–65%), 0.4 g of $NH_4Cl$, 0.1 g yeast extract, 0.5 g of $K_2HPO_4$, 0.1 g of $MgSO_4$, and 0.5 mL of trace element mixture; ferric citrate was added to obtain a final concentration of 60 μM. To make nitrate medium or ammonium/nitrate medium, 0.4 g of $NH_4Cl$ in the above recipe was replaced respectively with 1 g of $NaNO_3$ or 0.2 g of $NH_4Cl$ + 0.5 g $NaNO_3$. Synthetic wastewater medium was made as described previously [20] and sterilized at 121 °C for 30 min. MSR-1 cultures were incubated on a shaker (100 rpm, 30 °C) with MSR-1 antibiotics: nalidixic acid (Nx) 5 μg·$mL^{-1}$, ampicillin 5 μg·$mL^{-1}$, kanamycin sulfate (Km) 5 μg·$mL^{-1}$, and gentamicin (Gm) 5 μg·$mL^{-1}$. *Escherichia coli* cultures were incubated (shaking: 200 rpm, 37 °C) with *E. coli* antibiotics: ampicillin 100 μg·$mL^{-1}$, Km 100 μg·$mL^{-1}$, and Gm 20 μg·$mL^{-1}$.

### 2.2. Cell Growth and Magnetic Response

Optical density at a wavelength of 565 nm ($OD_{565}$) was measured every 2 h using a UV-visible spectrophotometer (model UNICO2100; Unico Instrument Co.; Shanghai, China). Magnetic response (Cmag) was determined as described previously [21]. Residual iron concentration was measured by ferrozine method in broth supernatant aspirated after centrifugation [22].

**Table 1.** Primers used in this study.

| Primer | Sequence (5′-3′) | Description |
| --- | --- | --- |
| Qrpoc-F | ATCTGGTCTACCGCCATTG | qRT-PCR for *rpoc* gene |
| Qrpoc-R | CCTTGCCGAACGAAATACC | qRT-PCR for *rpoc* gene |
| QmamA-F | GCCTATCCGTGGCGAAGAA | qRT-PCR for *mamA* gene |
| QmamA-R | TCGGCATCGTAAACCTGCT | qRT-PCR for *mamA* gene |
| QmamB-F | AGGTCGTGTGGTGGGCAT | qRT-PCR for *mamB* gene |
| QmamB-R | CGCTCATCCGCAGGCTTA | qRT-PCR for *mamB* gene |
| Qmms6-F | GGTTGGCGTTGGGAAGGT | qRT-PCR for *mms6* gene |
| Qmms6-R | CATCGCTCTGTGCCGCTT | qRT-PCR for *mms6* gene |
| QmmsF-F | TCGGGACGACGAGTTTGTC | qRT-PCR for *mmsF* gene |
| QmmsF-R | GGAACACCACGGAGACCAA | qRT-PCR for *mmsF* gene |
| QnapF-F | TGATGTCGCACAGCCTTAG | qRT-PCR for *napF* gene |
| QnapF-R | TGATGTCGCACAGCCTTAG | qRT-PCR for *napF* gene |
| QnirT-F | CCATTCACTACACCAACCGTTC | qRT-PCR for *nirT* gene |
| QnirT-R | ATGGCAGTTGCGGCATTC | qRT-PCR for *nirT* gene |
| QnorC-F | CGGTGTTCGTTGCCTTGA | qRT-PCR for *norC* gene |
| QnorC-R | CAGACATTGCCCAGTTCCG | qRT-PCR for *norC* gene |
| QnosZ-F | TCGCCACGGTGTCCTTT | qRT-PCR for *nosZ* gene |
| QnosZ-R | ATCACCTGACCGCTTTGGC | qRT-PCR for *nosZ* gene |
| Q*mg2046*-F | GCTCCATACCCAATGACGC | qRT-PCR for *mg2046* gene |
| Q*mg2046*-R | TGTCCACATCCTCGCCC | qRT-PCR for *mg2046* gene |
| Q*mgfnr*-F | GAGTTGAACCACGACGAAATCA | qRT-PCR for *mgfnr* gene |
| Q*mgfnr*-R | CGAACATCTCGCCCGAAA | qRT-PCR for *mgfnr* gene |
| *mgfnr*SF-EcoRI | CGGAATTCACCCTGACCGTGGGCAAGCCGGAA | Amplification upstream of *mgfnr* |
| *mgfnr*SR-SacI | CGAGCTCACCTTGTGATCGTCGTAATCC | Amplification upstream of *mgfnr* |
| *mgfnr*R-SacI | GCTGTTGTTCTTCCTGCT | Confirmation of *mgfnr* mutant |
| *mgfnr*F-SacI | TCCACCGAAATGAAACCG | Confirmation of *mgfnr* mutant |
| *mgfnr*XF-SacI | CGAGCTCTACCCAGTTGAAGCGTGAAG | Amplification downstream of *mgfnr* |
| *mgfnr*XR-XbaI | GCTCTAGAGAAATCGGAAAACAGCCCCA | Amplification downstream of *mgfnr* |
| P*mg2046*-F | GGAATTC ATGACGACGATGATCCA | Expression of *mg2046* protein |
| P*mg2046*-R | CCTCGAG TTAAACGTTCTCCCATC | Expression of *mg2046* protein |
| P*mgfnr*-F | CCGGAATTCGTGATCCCCATGCCGCC | Expression of *mgfnr* protein |
| P*mgfnr*-R | CCGCTCGAGCTAATGCGCCCCGCCGC | Expression of *mgfnr* protein |
| Erpoc-F | TGAAGGAAGCCAAGGACCT | EMSA for *rpoc* gene promoter |
| Erpoc-R | CGAGGGACGG GTCAAATCCC | EMSA for *rpoc* gene promoter |
| EnapF-F | CGGCGGTCAAGAAGATGA | EMSA for *nap* operon |
| EnapF-R | GAGTGCGCCCGAACAAGG | EMSA for *nap* operon |
| EnirT-F | AAGCAGCAGGGCGTTCCT | EMSA for *nir* operon |
| EnirT-R | AGTATTTTCATTTTGGACA | EMSA for *nir* operon |
| EnorC-F | AACCGATCTCATCGGCGAA | EMSA for *nor* operon |
| EnorC-R | TACCGCCATAAAAGATATT | EMSA for *nor* operon |
| EnosZ-F | AGACGTCGGGGCAGAAGGT | EMSA for *nos* operon |
| EnosZ-R | AGCGCGCCAAAGGACACCGT | EMSA for *nos* operon |
| E*mg2046*-F | TGTTGGGCGAAATCCTCGT | EMSA for *mg2046* gene promoter |
| E*mg2046*-R | GGCCAGGATG TCCACATCCT | EMSA for *mg2046* gene promoter |
| E*mgfnr*-F | TGGCTGAAATCTGCGAGGTT | EMSA for *mgfnr* gene promoter |
| E*mgfnr*-R | TCACGGGCGA CCTTGTGAT | EMSA for *mgfnr* gene promoter |

### 2.3. Construction of mgfnr Mutant Strain Δmgfnr

To construct *mgfnr* deletion mutant, an upstream fragment (1054 bp) was amplified using primer pair *mgfnr*S-F/*mgfnr*S-R (Table 1), and a downstream fragment (1207 bp) was amplified using primer pair *mgfnr*X-F/*mgfnr*X-R. These fragments were ligated into pMD18-T simple vector (TaKaRa Biotechnology; Dalian, China) for sequencing. Amplified upstream fragment was digested with EcoRI and SacI, and amplified downstream fragment was digested with SacI and XbaI. Gm cassette was digested from pUC-Gm vector with SacI. The above fragments were fused together into XbaI and BamHI sites of pUX19 vector to yield pUX-*mgfnr*. pUX- *mgfnr* was transformed into wild-type (WT).

MSR-1 by biparental conjugation using *E. coli* S17-1 as donor strain (Table 2). Colonies were screened and selected using Gm and Nx. Successful knockout of *mgfnr*, without deletion of magnetosome island-related genes, was confirmed by PCR.

**Table 2.** Strains used in this study.

| Strain or Plasmid | Description | Source or Reference |
|---|---|---|
| *E. coli* DH5α | *endA1 hsdR17* [r-m+] *supE44 thi-1 recA1 gyrA* [NalR] *relA relA1* Δ[*lacZYA-argF*] U169 *deoR* [Ø80Δ (*LacZ*) M15] | Novagen |
| *E. coli* DH5α-*mgfnr*S | DH5α containing pMD18-T-*mgfnr*S, Amp$^r$ | This study |
| *E. coli* DH5α-*mgfnr*X | DH5α containing pMD18-T-*mgfnr*X, Amp$^r$ | This study |
| *E. coli* S17-1 | *Thi endA recA hsdR* with RP4-2-Tc::Mu-Km::Tn7 integrated in chromosome; Sm$^r$, Tra | Novagen |
| *E. coli* S17-1-Δ*mgfnr* | S17-1 containing pUX-19-*mgfnr*S-Gm-*mgfnr*X, Km$^r$, Gm$^r$ | This study |
| *E. coli* BL21 (DE3) | F$^-$ *ompT hdS$_B$* (rB$^-$ mB$^-$) *gal dcm* (DE3), general purpose expression host | Novagen |
| *E. coli* BL21-P*mg2046* | BL21 containing pET28a (+)-P*mg2046*, Km$^r$ | This study |
| *E. coli* BL21-P*mgfnr* | BL21 containing pET28a (+)-P*mgfnr*, Km$^r$ | This study |
| MSR-1 WT | WT *M. gryphiswaldense*, Nx$^r$ | DSM 6361 |
| MSR-1 Δ*mg2046* | *mg2046*-deficient mutant, Nx$^r$, Gm$^r$ | This study |
| MSR-1 Δ*mgfnr* | *mg2046*-deficient mutant, Nx$^r$, Gm$^r$ | This study |

### 2.4. Transmission Electron Microscopic (TEM) Observation

Bacterial strains with 60 μM of ferric citrate were placed in SLM and incubated for 24 h at 30 °C. Cells were collected by centrifugation, washed twice with ddH$_2$O, resuspended, dropped onto a copper net, air-dried, and observed by TEM (model JEM-1230; JEOL; Tokyo, Japan). Number and size of magnetosomes were measured using ImageJ software program (NIH; Bethesda, MD, USA).

### 2.5. Quantitative Real-Time Reverse Transcription PCR (qRT-PCR)

WT and Δ*mg2046*/Δ*mgfnr* strains were cultured for periods of 6, 12, 18, and 24 h. RNA was extracted with TRIzol reagent (Tiangen; Beijing, China), and DNA was digested with DNase I (Takara; Shiga, Japan). RNA was reverse transcribed into cDNA using Moloney murine leukemia virus (M-MLV) reverse transcriptase (Promega; Madison, WI, USA). qRT-PCR was performed as per manufacturer's instructions, using an optical circulator 480 RT-PCR system and 480 SYBR Green I Master Kit (Roche; Mannheim, Germany). Reaction volume (20 μL total) contained 50 ng of template cDNA, 10 μL of SYBR Green I mix, and 0.5 mM of the particular primer. The housekeeping gene *rpoc* was used as the internal reference. Calculations were made using the $2^{-\Delta\Delta Ct}$ method, with triplicate samples.

### 2.6. Electrophoretic Mobility Shift Assays (EMSAs)

EMSAs were performed using digoxigenin gel shift kit (DIG) (2nd generation; Roche). Target DNA sequence was amplified and DIG-labeled in 20-μL reaction volume consisting of 0.4 nM of DIG-labeled probe, various concentrations of His$_6$-Mgfnr/His$_6$-Mg2046, 4 μL of bonding buffer, 1 μL of poly[d(I-C)], and ddH$_2$O. Specificity of Rok7B7/probe interaction was confirmed by adding 300x unlabeled rpoc (nonspecific probe) and specific probe to the reaction system. The sample was incubated for 30 min at 25 °C, PAGE was performed to separate protein-bound DNA from free DNA, gel transferred to positively-charged nylon membrane (Roche), and chemiluminescence signal detected using an imaging system.

### 2.7. Iron Absorption Capability

Iron concentration in medium was assayed using ferrozine [3-(pyridyl-2-yl)-5,6-bis(4-sulfophenyl)-1,2,4-triazine disodium salt]. Standard curves for the three strains were constructed based on various concentrations (0, 20, 40, 60, 80, 100 μM) of ferric citrate. One hundred microliters of ferric citrate was added with hydroxylammonium chloride (250 μL;

10% [$w/v$]), incubated for 5 min at room temperature, added with 20 µL of ferrozine (25 g·L$^{-1}$) and 630 µL of ddH$_2$O, and incubated for another 15 min. OD$_{565}$ was measured by spectrophotometry. At 6-h intervals, supernatants were obtained by centrifuging 1 mL of cell suspension (relative centrifugal force 8000; 3 min). Iron concentrations in supernatants were measured using the standard curves.

### 2.8. Protein Expression Vector Construction and Anaerobic Purification

Expression plasmids pET-28a-*mg2046* and pET-28a-*mgfnr* were constructed by PCR amplification using MSR-1 genome as template. *mg2046* (723 bp) and *mgfnr* (762 bp) gene fragments (upstream and downstream; containing EcoR I and Xho I restriction site sequences, respectively) were amplified, ligated into pMD18-T simple vector, and transformed into *E. coli* DH5α. Correctly sequenced strains were selected, strains containing pET-28a(+) plasmids were cultured, and plasmids were extracted and double digested with EcoR I and Xho I to recover target fragments and linear pET-28a(+) plasmids. Target fragments were combined with pET-28a(+), ligated, and transformed into *E. coli* BL21.

Constructed strains as above were cultured at 37 °C in 200 mL of LB broth added with 50 µg·mL$^{-1}$ of Km until OD$_{600}$ reached 0.6. Cells were placed in an ice bath for 18 min, and added with isopropyl β-D-1-thiogalactopyranoside (IPTG) to achieve a final concentration 1 mM of IPTG. *mg2046* was incubated at 150 rpm, at 16 °C, and *mgfnr* was incubated at 200 rpm and induced for 1 h at 37 °C. Methionine and ferric citrate were added to respective final concentrations of 25 and 200 µM, and incubation was continued. Cells were collected, disrupted in anaerobic environment by low-temperature centrifugation, and protein-containing supernatant was placed in nickel-nitrilotriacetic acid-agarose (Ni-NTA) column (GE Healthcare; Danderyd, Sweden), repeatedly vacuumed and exposed to nitrogen gas, transferred to anaerobic glovebox cabinet (MBRAUN LABstar; München, Germany), and eluted with buffers containing 20 mM Tris (pH 8.0), 300 mM NaCl, and imidazole at various concentrations. Purified proteins were stored in liquid nitrogen.

### 2.9. Chromatin Immunoprecipitation-Quantitative PCR (ChIP-qPCR)

Following anaerobic purification as above, proteins were sent out (BGI Genomics; Beijing, China) in order to prepare polyclonal antibodies for ChIP-qPCR. Purified Mg2046 and Mgfnr proteins were injected into rabbits to generate respective polyclonal antibodies. qPCR was performed using primer pairs listed in Table 1, and relative levels of Mg2046 and Mgfnr-precipitated DNA were determined by comparison with IgG control.

### 2.10. Total Nitrogen Content

Tested bacterial cells were cultured and harvested as in preceding sections. Total nitrogen content of cells was determined using a kit (cat # LH-NT-100; Lianhua Technology; Beijing, China) as per the manufacturer's instructions.

## 3. Results

### 3.1. Combined Effects of Dissimilatory Denitrification Pathway Genes Determined by EMSAs and ChIP-qPCR

Denitrifying effects of microorganisms in sewage are closely correlated with degree of denitrification gene expression. Two protein factors (Mg2046, MgFnr) were previously implicated in control of MSR-1 denitrification pathway [23]. We examined the capability of these factors to directly regulate specific denitrification genes, including *nap*, *nir*, *nor*, and *nos*.

His$_6$-labeled Mg2046 and Mgfnr proteins were expressed in *E. coli* BL21 and confirmed by western blotting (Figure S1). The capability of Mg2046 and Mgfnr to directly regulate genes involved in the dissimilatory denitrification pathway (NO$_3^-$ → NO$_2^-$ → NO → N$_2$O → N$_2$) was determined by EMSAs using anaerobically-purified His$_6$-Mgfnr and His$_6$-Mg2046. Both these tagged proteins combined with upstream sequences of *napF*, *nirT*, *nosZ*, and *mg2046* (Figure 1A,B), and His$_6$-Mg2046 combined with *norC* (Figure 1A), indicating

the capability of both Mg2046 and Mgfnr to directly regulate dissimilatory denitrification genes. Mgfnr directly regulated *mg2046*, whereas no direct regulator of *mgfnr* by Mg2046 was found.

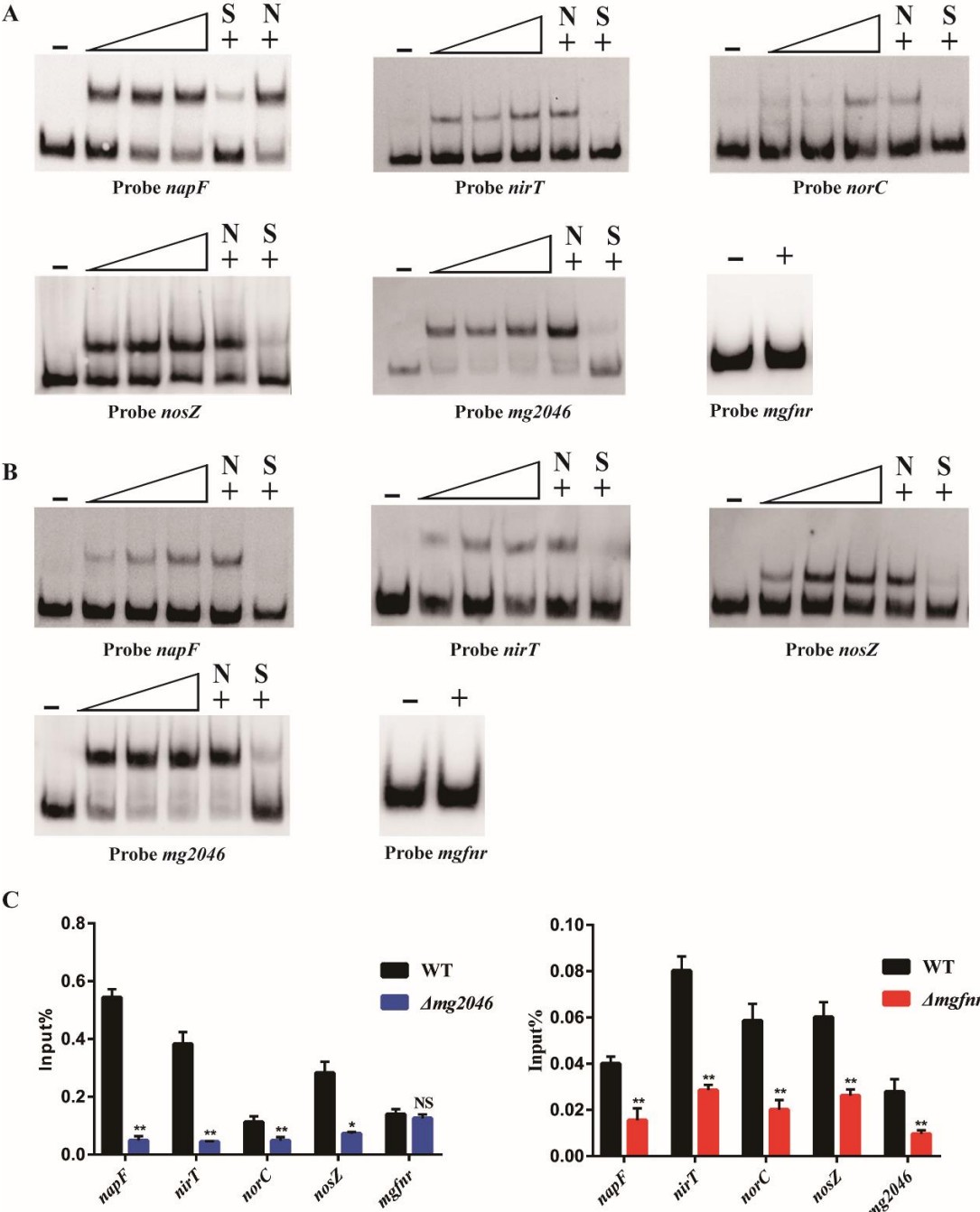

**Figure 1.** Mg2046 and Mgfnr combine with promoter regions of dissimilatory denitrification genes. (**A**) Mg2046 binds to promoter regions of *napF*, *nirT*, *norC*, *nosZ*, and *mg2046*. Each lane contains 0.4 nM of labeled probe. Lanes 1–6 respectively contain 0, 20, 40, 60, 60, and 60 ng of His$_6$-Mg2046. Lanes 5 and 6 are added with 100-fold unlabeled nonspecific competitor DNA (N) or unlabeled specific probe (S) to confirm binding specificity. (**B**) Mgfnr binds to promoter regions of *napF*, *nirT*, *nosZ*, and *mg2046*. Lanes 1–6 contain 0, 40, 80, 120, 120, and 120 ng of His$_6$-Mgfnr. (**C**) In vivo ChIP-qPCR analysis. MSR-1 strains WT, Δ*mg2046*, and Δ*mgfnr* were immunized with anti-Mg2046 and -Mgfnr antibodies for 24 h. y-axis: relative abundance of *mg2046* and *mgfnr* for each point. NS: no significant difference (*t*-test). * $p < 0.05$; ** $p < 0.01$. Error bar: SD from three replicate experiments.

Combination of Mgfnr and Mg2046 with dissimilatory denitrification gene promoters in vivo was further examined by ChIP-qPCR using WT, Δ*mgfnr* (constructed in this study; confirmed by PCR) (Figure S2), and Δ*mg2046* (constructed by Wang et al. [24]). WT strain was used as negative control, qRT-PCR analysis of product DNA showed that no enrichment of Mg2046 on *mgfnr* was detected; enrichment levels of Mgfnr and Mg2046 on *napF*, *nirT*, *norC*, *nosZ* and *mg2046* were at least 1-fold higher in Δ*mg2046*/Δ*mgfnr* than that in control in all samples immunoprecipitated (Figure 1C). These findings indicate that Mg2046 regulates dissimilatory denitrification genes and itself, but does not directly regulate *mgfnr*, whereas Mgfnr regulates the dissimilatory denitrification genes and also *mg2046*.

### 3.2. Transcription Levels of Dissimilatory Denitrification Pathway Genes

Variations of dissimilatory denitrification pathway in Δ*mg2046* and Δ*mgfnr* were evaluated by qRT-PCR analysis of key genes (*napF*, *nirT*, *norC*, *nosZ*). Elimination of *nap* and *nir* impairs magnetite biomineralization, resulting in fewer, smaller, and more irregular crystals during the denitrification process. Microaerobic respiration may interfere with magnetosome synthesis by disrupting proper oxide balance [19,25]. Δ*mg2046*, relative to WT, did not show significant difference of *nirT* or *norC* transcription levels at 6 or 12 h, nor of *napF* or *nosZ* levels at 6 h. Levels of *napF*, *nirT*, *norC*, and *nosZ* decreased significantly in Δ*mg2046* relative to WT during 18–24 h but increased significantly in Δ*mgfnr* (Figure 2). Levels of dissimilatory denitrification genes were relatively low from 6–12 h. Consumption of oxygen in medium from 18–24 h resulted in increased levels of dissolved oxygen and of dissimilatory denitrification genes. Gene transcription was significantly higher in WT than in Δ*mg2046*, indicating a positive regulatory effect of *mg2046* on denitrification pathway as oxygen content declined. Transcription levels of *mgfnr* genes did not differ significantly between Δ*mg2046* and WT, whereas levels of *mg2046* genes were significantly lower (Figure 2E,F), indicating a unidirectional positive regulatory effect of *mgfnr* on *mg2046*.

Energy-saving wastewater treatment plants have received increasing research attention [26]. Microaerobic wastewater treatment has the potential to reduce energy consumption and improve operational efficiency [27,28]. Required DO concentration in microaerobic reactors is as low as 0.3–1.0 mg·L$^{-1}$ and can save aeration energy [29]. *mgfnr* genes had a negative regulatory effect on the pathway during all growth periods, suggesting that Δ*mgfnr* mutation could potentially increase nitrate utilization efficiency. We utilized a microaerophilic MTB strain, MSR-1, to remove nitrate from synthetic wastewater medium. Δ*mgfnr* is a promising candidate for industrial nitrate removal from wastewater.

### 3.3. Phenotypic Analysis of WT and Δ*mg2046*/Δ*mgfnr*

Large quantities of activated sludge (an umbrella term for microbial communities and the organic and inorganic substances on which they depend [30]) are typically produced during sewage treatment. During denitrification processes, sludge is distributed throughout the sewage system and needs to be collected for treatment. Ease of solid/liquid separation is directly related to treatment efficiency of activated sludge. The activated sludge with MTB can be quickly separated from the processing wastewater under an added magnetic field, which presents a unique advantage in sewage treatment. An important consideration in studying the denitrification power of mutant strains is that their magnetosome synthesis capability be fairly unaffected in order to facilitate application of magnetic adsorption.

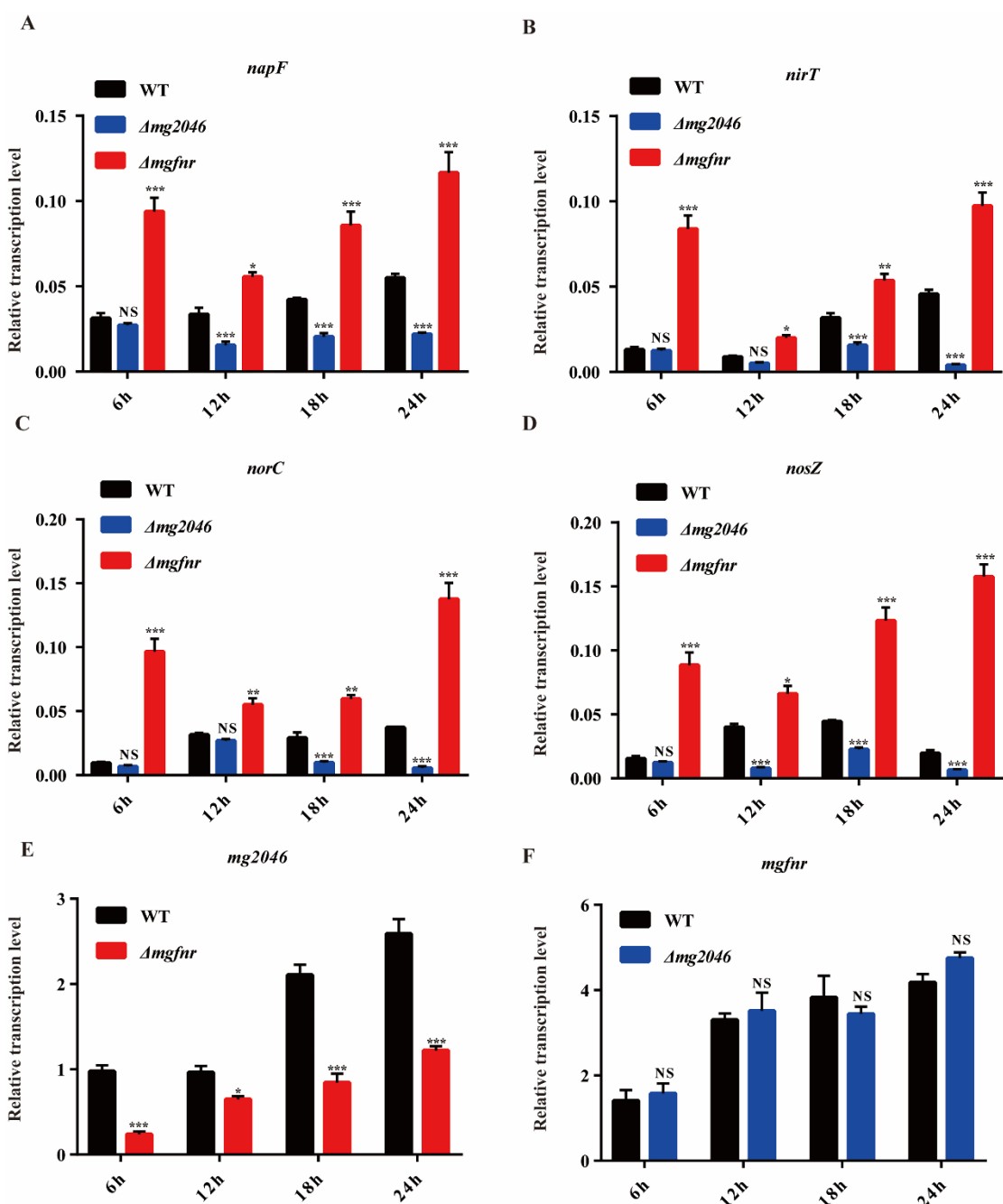

**Figure 2.** Relative transcription levels of dissimilatory denitrification genes in WT, Δ*mg2046*, and Δ*mgfnr*. (**A–F**): *napF*, *nirT*, *norC*, *nosZ*, *mg2046*, and *mgfnr* genes. Reference gene: *rpoC*. NS: no significant difference (*t*-test). * $p < 0.05$; ** $p < 0.01$; *** $p < 0.001$. Error bar: SD from three replicate experiments.

WT and Δ*mgfnr* MSR-1 strains were observed by TEM, and magnetosome numbers and particle sizes were quantified. In WT, magnetosomes were densely arranged and had uniform particle size. In Δ*mg2046*, magnetosomes were less dense and had shorter chains. Δ*mgfnr* showed greater synthesis of irregular magnetosomes, including some with small size (Figure 3A). Relative to WT, magnetosome number and particle size were significantly smaller for Δ*mg2046*, but similar for Δ*mgfnr* (Figure 3B).

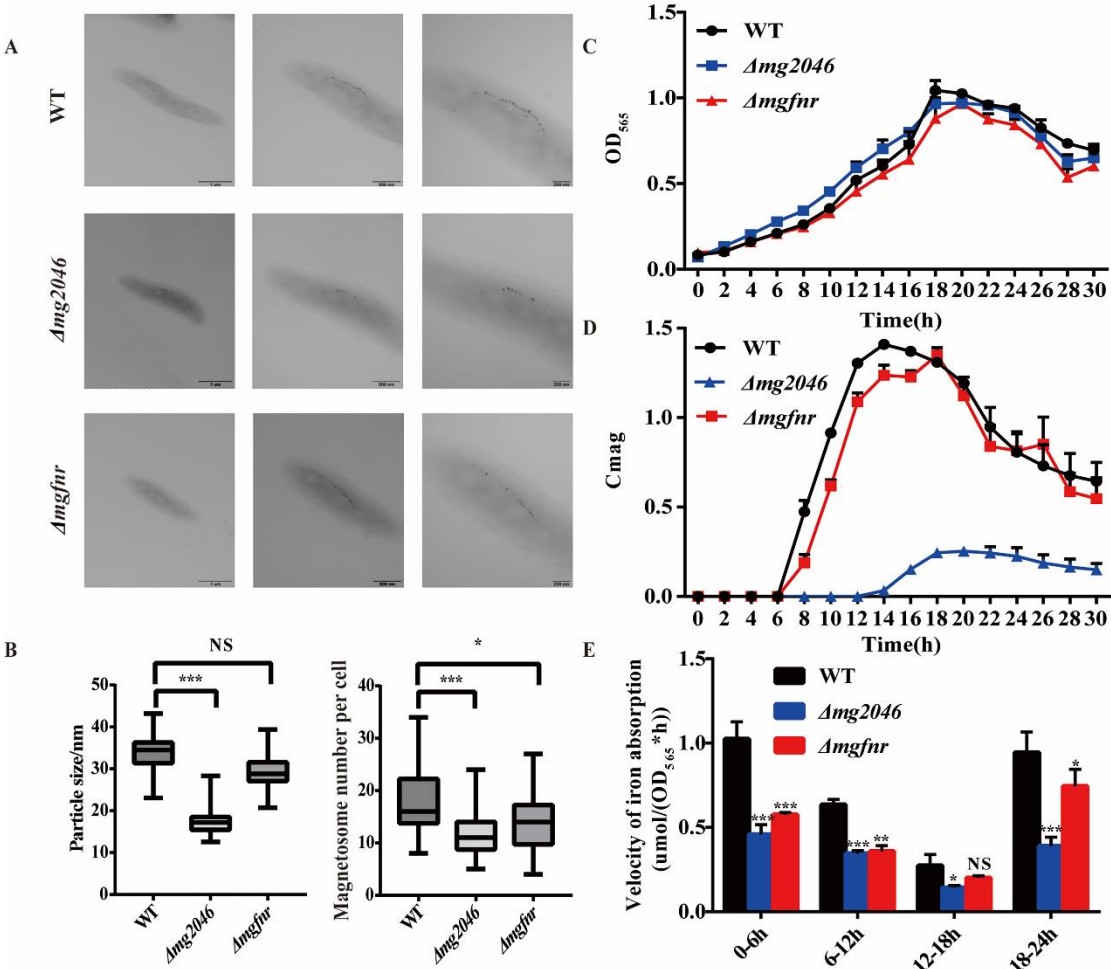

**Figure 3.** Phenotypic analyses of WT, Δ*mg2046*, and Δ*mgfnr*. (**A**) TEM images with progressive enlargement from left to right. Scale bars: 1 μm, 500 nm, 200 nm. (**B**) Magnetosome particle size and number per cell. (**C**) Cell growth curve ($OD_{565}$). (**D**) Magnetic response (Cmag) curve. (**E**) Cellular iron absorption rate. Data shown represent mean ± SD from three replicates. NS: no significant difference (*t*-test). * $p < 0.05$; ** $p < 0.01$; *** $p < 0.001$. Error bar: SD from three replicate experiments.

MTB were cultured in a thermostatic shaker. For construction of the Cmag curve, vertical and horizontal magnetic fields were added by modifying the spectrophotometer, measuring absorbance, and calculating Cmag values to reflect differences in number and maturity of magnetosomes. $OD_{565}$ growth curves of Δ*mgfnr*/Δ*mg2046* were similar to that of WT (Figure 3C). Comparison of Cmag curves showed that magnetosome synthesis in Δ*mg2046* began at 12 h and that Cmag value was significantly lower than that of WT, indicating slower magnetosome synthesis and reduced magnetic response. The magnetic response curve of WT was similar to that of Δ*mgfnr*, indicating that absence of *mgfnr* did not strongly affect magnetosome synthesis (Figure 3D). Iron absorption rates were calculated at 6-h intervals, based on measurement of residual iron content in cells. Rates for Δ*mg2046*/Δ*mgfnr* were lower than for WT over various time periods. Rates for Δ*mgfnr* were higher than those for Δ*mg2046* during the 0–6 h and 18–24 h periods, consistent with the higher Cmag value of Δ*mgfnr* (Figure 3E).

### 3.4. Transcription of Genes Involved in Magnetosome Synthesis

The phenotypic analyses clearly showed that magnetosome synthesis was altered by mutations of *mg2046* and *mgfnr*. Changes in gene transcription levels of magnetosome islands (MAI) were evaluated by examining key MAI genes *mamA*, *mamB*, *mamF*, and *mamY*. The *mamAB* gene cluster plays a key role in magnetosome synthesis [31]. Transcription levels of the four genes during various periods were significantly higher for WT than for *Δmg2046*, consistent with the reduced magnetosome synthesis and magnetic response of *Δmg2046* during cell growth (Figure 4A,B). Transcription levels of the four genes during the most crucial magnetosome synthesis period (6–12 h) did not differ significantly between *Δmgfnr* and WT. During the 18–24 h period, *mamA*, *mamB*, and *mamY* levels were significantly lower for *Δmgfnr* than for WT (Figure 4C,D). These findings suggest that magnetosome synthesis in *Δmg2046* was blocked, and abnormal crystals were generated during early cell growth stages because of reduced expression of the four genes. Gene transcription levels for *Δmgfnr* during the crucial 6–12 h period did not differ notably from those for WT, indicating that this mutation had no major effect on magnetosome synthesis and will not affect magnetic adsorption sedimentation. In contrast, normal functioning of *mg2046* plays an important role in magnetosome synthesis through direct or indirect positive regulatory effects.

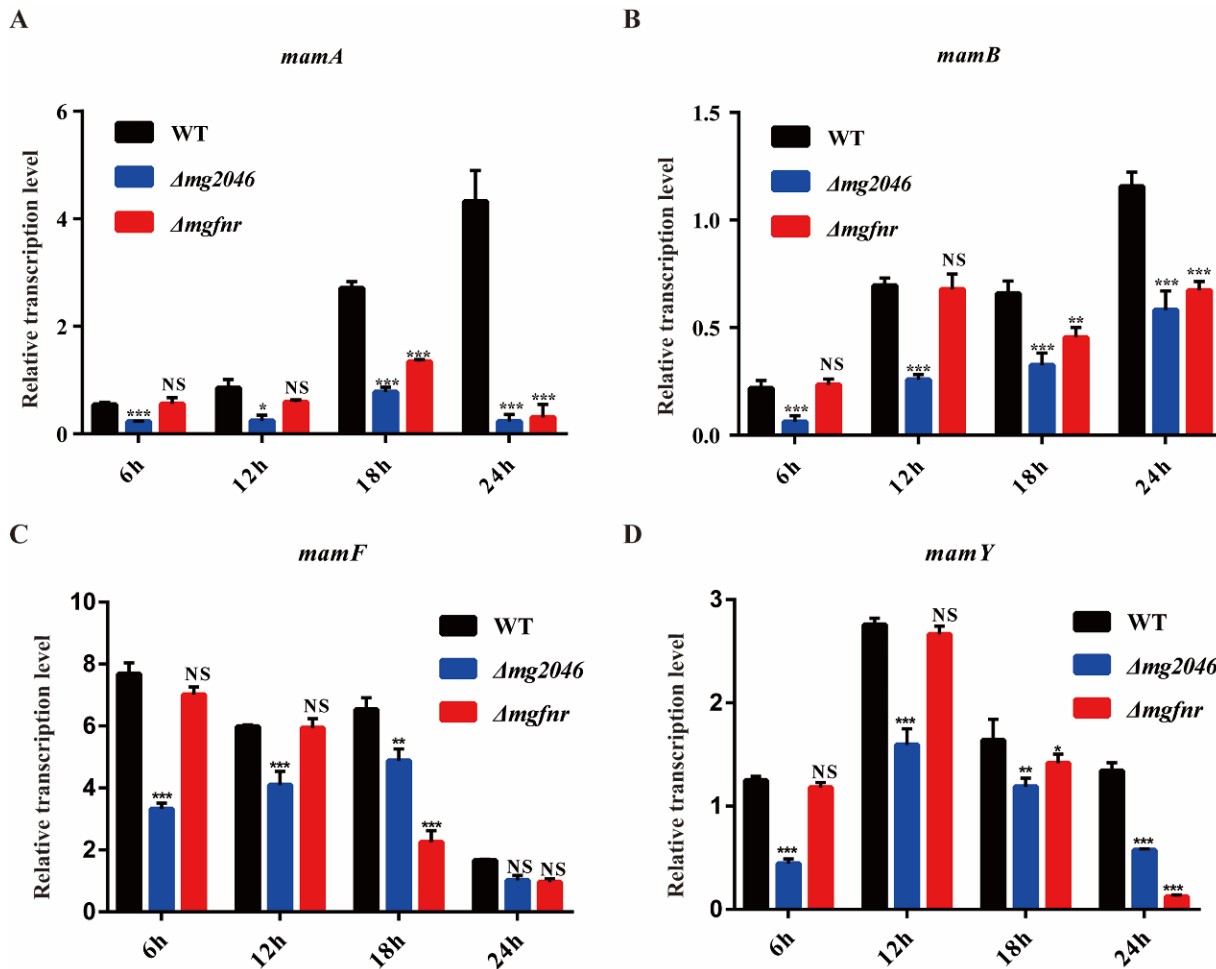

**Figure 4.** Relative transcription levels of MAI genes in WT, *Δmg2046*, and *Δmgfnr*. (**A–D**): *mamA*, *mamB*, *mamF*, and *mamY* genes. Reference gene: *rpoC*. NS: no significant difference (*t*-test). * $p < 0.05$; ** $p < 0.01$; *** $p < 0.001$. Error bar: SD from three replicate experiments.

### 3.5. Comparative Utilization of Various Nitrogen Sources by WT and $\Delta mg2046/\Delta mgfnr$

Absorption and utilization capabilities of WT and $\Delta mg2046/\Delta mgfnr$ for various nitrogen sources (ammonium [$NH_4^+$], nitrate [$NO_3^-$], ammonium/nitrate [$NH_4^+/NO_3^-$]) were evaluated. In $NH_4^+$ medium, total nitrogen removal rates after 12 h culture for WT, $\Delta mg2046$, and $\Delta mgfnr$ were respectively 0.363, 0.178, and 0.393 mmol·L$^{-1}$·h$^{-1}$, and total nitrogen content in medium did not differ notably for WT vs. $\Delta mgfnr$ (Figure 5A,B). Total nitrogen content in medium, relative to WT, was significantly higher for $\Delta mg2046$. These findings suggest that $NH_4^+$ absorption and utilization capabilities were similar for WT and $\Delta mgfnr$ but lower for $\Delta mg2046$.

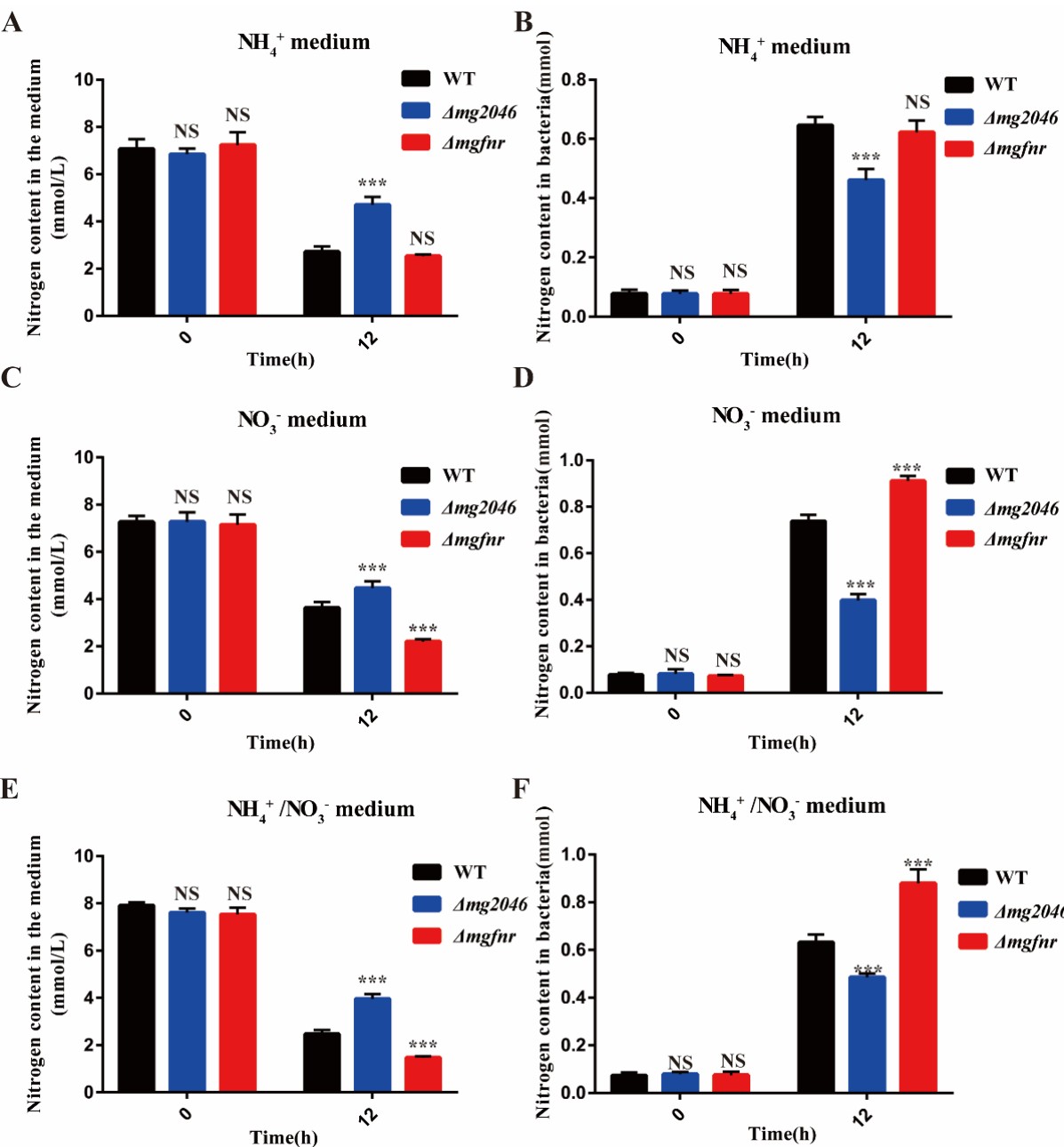

**Figure 5.** Total nitrogen content of media and bacteria for WT, $\Delta mg2046$, and $\Delta mgfnr$. (**A**) $NH_4^+$ medium. (**B**) Bacteria in $NH_4^+$ medium. (**C**) $NO_3^-$ medium. (**D**) Bacteria in $NO_3^-$ medium. (**E**) $NH_4^+/NO_3^-$ medium. (**F**) Bacteria in $NH_4^+/NO_3^-$ medium. NS: no significant difference (*t*-test). *** $p < 0.001$. Error bar: SD from three replicate experiments.

Total nitrogen removal rates after 12 h culture for WT and $\Delta mg2046/\Delta mgfnr$ were respectively 0.302, 0.234, and 0.412 mmol·L$^{-1}$·h$^{-1}$ in NO$_3^-$ medium, and 0.454, 0.304, and 0.505 mmol·L$^{-1}$·h$^{-1}$ in NH$_4^+$/NO$_3^-$ medium (Figure 5C–F). Total nitrogen content in medium, relative to WT, was significantly higher for $\Delta mg2046$ but lower for $\Delta mgfnr$, and nitrogen removal rates were respectively 69.20%, 65.09%, and 80.50%. Total nitrogen content in bacteria, relative to WT, was significantly lower for $\Delta mg2046$ but higher for $\Delta mgfnr$. These findings indicate that NO$_3^-$ and NH$_4^+$/NO$_3^-$ absorption and utilization capabilities of $\Delta mgfnr$, relative to WT, are significantly higher, and it has superior pollution treatment capacity for major nitrogen-source nitrates in the environment. $\Delta mgfnr$ can use NO$_3^-$ and NH$_4^+$ simultaneously as nitrogen sources and adapt to a variety of sewage environments.

*3.6. Comparative Utilization of Nitrogen Sources from Synthetic Wastewater by WT and $\Delta mg2046/\Delta mgfnr$*

The three bacterial strains were cultured in synthetic wastewater to evaluate feasibility of using MTB for treatment of industrial wastewater from the stainless steel industry. Samples were taken at 0, 12, and 24 h for determination of total nitrogen content in medium and in bacteria. Total nitrogen levels did not differ notably at 0 h. Total nitrogen removal rates in medium for WT, $\Delta mg2046$, and $\Delta mgfnr$ were respectively 0.362, 0.302, and 0.456 mmol·L$^{-1}$·h$^{-1}$ at 0–12 h, and 0.210, 0.205, and 0.155 mmol·L$^{-1}$·h$^{-1}$ at 12–24 h (Figure 6A). Total nitrogen content in bacteria generally increased with culture time, and relative to WT was higher for $\Delta mgfnr$ and lower for $\Delta mg2046$ (Figure 6B). Thus, denitrification efficiency in synthetic wastewater declined for $\Delta mg2046$ but increased for $\Delta mgfnr$ (0.456 mmol·L$^{-1}$·h$^{-1}$ during 0–12 h; 0.155 mmol·L$^{-1}$·h$^{-1}$ during 12–24 h). Total nitrogen removal rates for $\Delta mgfnr$ were 64.09% at 12 h and 85.83% at 24 h. $\Delta mgfnr$ is potentially useful for nitrate processing in synthetic industrial wastewater from the stainless steel industry, with a variety of applications.

Feasibility of magnetic adsorption by the three MTB strains was evaluated by placing cultures above a strong magnet and measuring OD$_{565}$; cultures without a magnet were used as controls. Sedimentation rates of the three control groups were similar (OD$_{565}$ decreased from 1.0 at 0 h to 0.3 at 148 h); the rates declined rapidly from 12 to 96 h but showed minimal change from 96 to148 h (Figure 6C). Magnetic adsorption rates were significantly higher for experimental groups than control groups (Figure 6D). OD$_{565}$ for experimental groups declined from 1.0 at 0 h to 0.15–0.3 at 24 h, whereas OD$_{565}$ for control groups was ~0.7 at 24 h. OD$_{565}$ for WT and $\Delta mgfnr$ was near 0 at 72 h, whereas that for $\Delta mg2046$ approached 0 at 96 h, presumably because magnetosome synthesis was blocked. These findings suggest that magnetic adsorption enhances sedimentation rates for MTB, reduces energy consumption required for separation of activated sludge, and has a wide variety of potential applications.

The traditional activated sludge system for wastewater treatment has always been the object of a lot of research [32,33]. Wastewater treatments using current activated sludge process (ASP) inevitably produce waste biomass. The water treatment industry has developed rapidly during the past decade, with associated increase in amount of activated sludge produced by sewage processing [34]. Various pretreatment methods for active sludge (including chemical, mechanical, biological, and thermal hydrolysis methods) have been developed [35,36]. For all of these, improved separation of liquid and solid components will result in greater energy efficiency, better sludge dewater ability, and reduced sludge viscosity, making the method more feasible and practicable [37]. The fact that MTB can be easily separated from wastewater using magnetic fields [18] is a unique feature that makes $\Delta mgfnr$ promising candidates for practical application in sludge pretreatment.

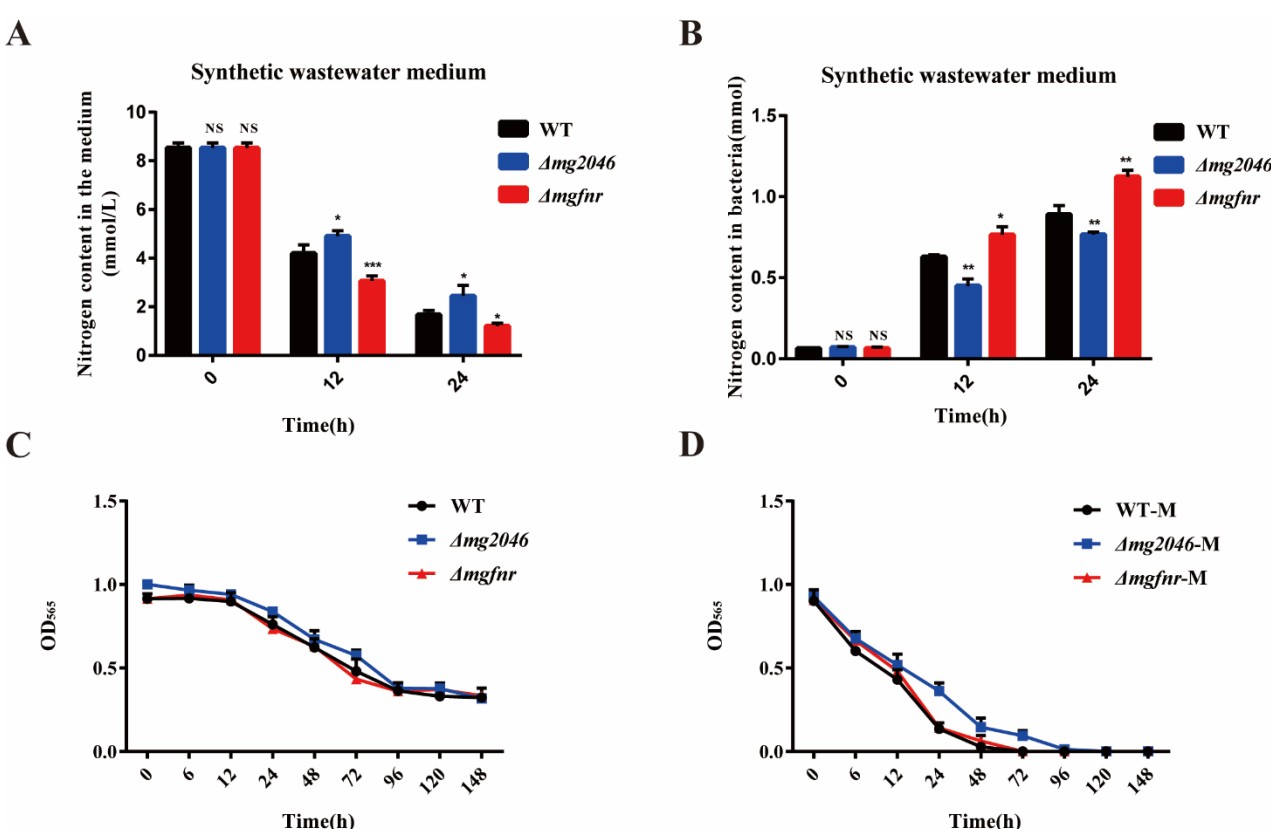

**Figure 6.** Total nitrogen content of media and bacteria in synthetic wastewater medium for WT, Δ*mg2046*, and Δ*mgfnr* (**A**,**B**), and OD$_{565}$ curves (**C**,**D**). (**A**) Media. (**B**) Bacteria. (**C**) OD$_{565}$ curve during natural sedimentation. (**D**) OD$_{565}$ curve during magnetic adsorption sedimentation. NS: no significant difference (*t*-test). * $p < 0.05$; ** $p < 0.01$; *** $p < 0.001$. Error bar: SD from three replicate experiments.

## 4. Conclusions

Proteins Mg2046 and Mgfnr were shown to directly regulate genes involved in the dissimilatory denitrification pathway. *M. gryphiswaldense* MSR-1 deletion mutant strains Δ*mg2046* and Δ*mgfnr* were constructed, and their functions and relationships were evaluated based on phenotypic and genetic characteristics. *mgfnr* mutation did not notably affect magnetosome synthesis, and transcription levels of dissimilatory denitrification genes for Δ*mgfnr* were higher than those for WT. Total nitrogen removal rates in $NH_4^+$, $NO_3^-$, $NH_4^+/NO_3^-$, and synthetic wastewater medium were significantly higher for Δ*mgfnr* than for WT or Δ*mg2046*. Magnetic adsorption was shown experimentally to significantly increase the sedimentation rate of activated sludge for MTB.

**Supplementary Materials:** The following supporting information can be downloaded at: https://www.mdpi.com/article/10.3390/pr10030591/s1, Figure S1: Protein purification and verification. (A) SDS-PAGE purification of proteins His6-Mgfnr and His6-Mg2046. (B) Confirmation of target protein by Western blotting; Figure S2: Construction of Δ*mgfnr* strains and confirmation by PCR. (A) Construction of Δ*mgfnr* strains (schematic). (B) Confirmation by PCR of Δ*mgfnr* strains. Markers: 3000, 2000, 1200, 800, 500, and 200 bp. (C) Detection by PCR of mam genes in mutant strains.

**Author Contributions:** H.Z.: conceptualization, methodology, software, formal analysis, investigation, data curation, writing—original draft, visualization; B.P.: methodology, software, formal analysis, Writing—review & editing, visualization; S.L.: investigation, formal analysis, software, formal analysis; S.M.: investigation, methodology; J.X.: investigation, data curation; Y.W.: Writing—review & editing, supervision, funding acquisition; J.T.: conceptualization, methodology, writing—review &

editing, supervision, project administration, funding acquisition. All authors have read and agreed to the published version of the manuscript.

**Funding:** This research received no external funding.

**Acknowledgments:** This study was supported by Key Project of Inter-Governmental International Scientific and Technological Innovation Cooperation (2019YFE0115800), the National Natural Science Foundation of China (No. 31570037, 21577170), the Project for Extramural Scientists of State Key Laboratory of Agro-biotechnology (2020SKLAB6-6), Science and Technology Power Economy 2020 Key Projects (SQ2020YFF0401976), and Beijing Municipal Science and Technology Program (Z201100007920010).The authors are also grateful to S. Anderson and Tara Penner for English editing of the manuscript.

**Conflicts of Interest:** The authors declare no conflict of interest.

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
