# Peer review of "Construction of Recombinant Magnetospirillum Strains for Nitrate Removal from Wastewater Based on Magnetic Adsorption"

_processes, doi:10.3390/pr10030591_

Round 1

Reviewer 1 Report

Abstract and introduction: (both) the first line, the phrase says "they are one of the main causes" I think it is absolutist so I suggest changing it to "It is one of the main causes of..." 2.10. Total nitrogen content: is the nitrogen determination total (TNT) or are there two kits, one for ammonium and one for nitrate? Nitrite? In the graph (fig. 5) the determinations are expressed in NO3- and NH4+ In fig. 5 A The bars show mmol/l and in B in mg/l. I Would it be possible to express the determinations in N-NH4 - N-NO3 or in moles?

also in fig. 6. In addition to the WT is there a control – no bacteria?

Author Response

I would like to thank you for your careful reading, helpful comments, and constructive suggestions, which has significantly improved the presentation of our manuscript. Here are my responses to your suggestions:

Point 1:  the first line, the phrase says "they are one of the main causes" I think it is absolutist so I suggest changing it to "It is one of the main causes of..." 

Response 1: The text has been revised according to the reviewer's comments, and “Nitrate ions (NO3) in wastewater are a major cause of” in line 18 has been changed to “Nitrate ion (NO3) in wastewater is one of the main cause of”.

Point 2: 2.10. Total nitrogen content: is the nitrogen determination total (TNT) or are there two kits, one for ammonium and one for nitrate? Nitrite?

Response 2: There was only one kit in 2.10 which converted all nitrogen sources to nitrate and then determined the total nitrogen content.

Point 3: In the graph (fig. 5) the determinations are expressed in NO3- and NH4+ In fig. 5 A The bars show mmol/l and in B in mg/l. I Would it be possible to express the determinations in N-NH4 - N-NO3 or in moles?also in fig. 6.

Response 3:  The unit “mg” in pictures 5B, D, F and Fig. 6B has been replaced with  “mmol”, and the unit “mg/L” in Fig. 6A has been replaced with “mmol/L”; Then recalculate based on unit changes in the figure,line 23 “Δmgfnr (28.26 mg·L−1·h−1) than for wild-type (22.42 mg·L−1·h−1)” was changed to “Δmgfnr (0.456 mmol·L−1·h−1) than for wild-type (0.362 mmol·L−1·h−1)”, line 282 “respectively 22.50, 11.05, and 24.34 mg·L−1·h−1” was changed to “respectively 0.363, 0.178, and 0.393 mmol ·L−1·h−1, line 288 ”respectively 18.75, 14.52, and 25.55 mg·L−1·h−1in NO3 medium, and 28.12, 18.84, and 31.33 mg·L−1·h−1 ” was changed to “respectively 0.302, 0.234, and 0.412 mmol·L−1·h−1 in NO3 medium, and 0.454, 0.304, and 0.505 mmol·L−1·h−1 ", line 303 "respectively 22.42, 18.73, and 28.26 mg·L−1·h−1 at 0-12 h, and 13.02, 12.72, and 9.59 mg·L−1·h−1 ” was changed to “respectively 0.362, 0.302, and 0.456 mmol·L−1·h−1 at 0-12 h, and 0.210, 0.205, and 0.155 mmol L−1·h−1”, line 307 “Δmgfnr (28.26 mg L−1·h−1 during 0–12 h; 9.59 mg ·L−1·h−1” was changed to “Δmgfnr (0.456 mmol·L−1·h−1 during 0-12 h; 0.155 mmol·L−1·h−1 ”.

Point 4:  In addition to the WT is there a control – no bacteria?

Response 4: All cultures are carried out under sterile conditions, so the initial state at 0 h is close to the nitrogen content of the sterile medium (no bacteria).

Once again, thank you very much for your comments and suggestions.

Reviewer 2 Report

I accept, with minor revisions.

Author Response

We have re-checked the manuscript and sincerely thank the reviewers for their enthusiastic work.